# Modification Tests to Optimize Highway Construction in Crown of Slate Random Embankments with Compaction Quality Control

**DOI:** 10.3390/ma13051139

**Published:** 2020-03-04

**Authors:** Evelio Teijón-López-Zuazo, Ángel Vega-Zamanillo, Miguel Ángel Calzada-Pérez, Luis Damián Ramos-Pereira

**Affiliations:** 1Construction and Agronomy Department, Zamora Polytechnical School, Viriato Campus, University of Salamanca, 49022 Zamora, Spain; ldramos@usal.es; 2Department of Transportation and Projects and Processes Technology, Civil Engineering Technical School of Santander, University of Cantabria, Los Castros Avenue, 39005 Santander, Spain; vegaa@unican.es (Á.V.-Z.); calzadam@unican.es (M.Á.C.-P.)

**Keywords:** slate, crown, random fill, compaction quality control, wheel-tracking test, topographic settlement

## Abstract

The compaction control of random fills has developed very little due to the limitations of particle sizes, with methods usually using a simple procedural control. In order to develop new quality control procedures for random slate filling, the necessary field work and laboratory tests were carried out. New test procedures for wheel-tracking or settlement tests have been successfully investigated. A total of 4500 in situ measurements of density and 580 topographic settlements, 960 wheel-tracking trolley tests and more than 130 slab rolling tests have been determined. An analysis of variance (ANOVA) has been carried out, obtaining correlations between tests in order to replace the deductibles. The variables that were used to analyze variance were the average lot density, the average settlement between last and first roller pass, the average wheel impression after test carriage, the first vertical modulus of the plate bearing test (PLT), the second vertical modulus of the PLT and the relation between second and first modulus of the PLT. The research proposes a new procedure for the quality control of the compaction of the randomized slate filling used in the crown area.

## 1. Introduction

In highway infrastructures, quality control of random embankments is carried out using tests that cannot correctly evaluate the compaction process [1]. Sopeña [2] indicated that topographic control has no reference values.

Zhong et al. [3] developed an automatic process for the monitoring of compaction parameters. This avoids the influence of the operator or the limitations of conventional methods. The process was correctly applied to the Nuozhadu dam in China.

For Teijón el al. [1], pit gradings weighing rock fractions were ineffective. Well-track tests ran under normal compaction conditions. The plate load test (PLT) requires the diameter of the load element to be five times the maximum size of the aggregate. Radioactive isotope density testing is conditioned by particle sizes and layer thicknesses less than 30 cm. The modified Proctor has the disadvantage of replacing sizes larger than 20 mm, with a minimum of 70% substituted material. The sand method is also useless as it is limited to a maximum size of 50 mm.

Mazari and Nazari [4] consider that quality and density are not related. Density is a control element; quality must be based on determinations of the modulus of elasticity for quality acceptance. This can be estimated by means of formulas or usual values.

For Fernández et al. [5], the limited development of compaction control tests justifies the execution of test sections.

A summary about the method of embankment quality control tests for a better comprehension, with the research deficiencies based on the literature review, is described in Table 1. It shows the research gaps that the authors identified.

This study focuses on slate rocks. The reason for choosing the slate family is that these rocks are usually obtained from highway diggings, similar to those obtained from the demolition of buildings or pavements. Thus, the scientific knowledge was focused on slates because is a rock widely abundant and therefore the results on this material have wide applicability.

Possible correlations between the wheel-tracking test, topographic settlement, PLT and the on-site density test have been analyzed. The statistical analysis provided dependency relationships, allowing a new method of compaction control to be defined by applying only representative tests. It avoids unnecessary interruptions and the need for expensive equipment. It also proposes values that allow an efficient compaction control.

This research establishes the revision of certain methods, such as wheel-tracking or topographic settlement tests, in slate random embankments for highways, particularizing the proposed new compaction control method to be used on slate rocks in the crown zone with a maximum layer thickness of 600 millimeters.

Fernández et al. [5] considered that it is possible to use rocks with low resistances (below 25 MPa) to obtain random fill. Such rocks are usually obtained from the demolition of structural and pavement concrete. Although the quality of slates and shales is lower than that of other materials such as greywackes, their strong anisotropic behavior associated with stratification and granulometric degradation after compaction is difficult to predict. Even so, they can be considered stable rocks and suitable for usage as random fill. In general, this can encourage high percentages of slab forms. Several experiments on test sections of stony materials have been conducted. Since the mechanical strength of grauwacke (a detrital rock formed from the consolidation of disintegrated granite minerals) is lower than that of granite, it usually ends up forming random fill.

The laboratory and field compaction data reported by Horpibulsuk et al. [6] show that the relationship between relative density and the number of roller passes is represented by the logarithm function in laterite soils. Likewise, Oteo [7] associated the requirements of materials to be used in fills with granulometry and density.

The effect of size ratio and air volume between the particles on the aggregate structure and packing density of binary mixtures was researched by Pouranian, M. R. and Haddock, J.E. [8]. In addition, compaction parameters, including compaction slope, initial density, locking point and compaction energy index have been analyzed.

Onana et al. [9] characterized the charnockite of Cameroon. The samples presented characterization tests with fine contents between 16 and 44%, high plasticity rates 26%–55%. In terms of its mechanical properties, it presents a high bearing capacity, with CBR 31-68 indexes, average RCS values 0.88–1.20 MPa and low tensile strength 0.07–0.15 MPa.

According to the Casagrande plasticity chart, the tested laterite gravels are clayey and highly plastic, which is due to their high kaolinite content. Southern Cameroon laterite materials are very low compressibility clayey gravel (GC) or silty gravel (GM) and can be used as sub-base layers for any volume of traffic.

Regarding embankment seats, Sagaseta [10] indicated that associations could develop in random fills, which are made up of evolutionary materials, such as shales (fine grain detrital sedimentary rocks). In these cases, deferred settlements can become increased by the action of external agents (weathering, freezing cycles) that highly damage these rocks.

The Construction Embankment Technical Guide [11] provides a classification of rocks. The R_6_ group includes metamorphic rocks, such as slates and schists. The working method should be defined for the available machinery, earth moving methods, layer thickness, compaction procedures, number of roller passes, adjustment to optimum moisture and similar tests.

Oteo [7] considered altered granite as a random fill. A specific study should be conducted before its excavation, transport and setting in place, and the appropriate control system must also be selected, since the classic Proctor test is hardly useful as a reference for such heterogeneous materials. For control of compacted random fill, the plastic density method, alongside geophysical methods, would be best. Radioactive isotope density can lead to specific problems in rock lacking fine fractions, since, because of their dimensions, the particles of such rocks do not allow the introduction of a gamma emitter into the ground. While it is still possible to measure backscattering, the soil volume tested for influence is inevitably smaller. Therefore, the measurements performed belong to the most superficial area, which is where the impact of the compaction energy is higher.

Wan-Huan et al. [12] estimated the soil–water characteristic curve (SWCC) of soils with different initial dry densities.

Based on several experiments using highway test sections, Fernández et al. [5] concluded that the results obtained from plate bearing tests show scatter.

For the wheel-tracking test, seat measurements are performed before and after carriage passes at ten points that are 1m apart from each other.

Sun et al. [13] carried out certain experiments on 75 × 75 × 87 cm crushed rock samples subjected to vertical cyclic loading. Three coarsely crushed rock samples with initial grain sizes of 16–40, 25–50 and 50–80 mm were used to measure the corresponding parameters.

Garcia et al. [14] analyzed the granular sub-base of the railway. Thus, Ev_2_ is not associated with compaction, using Ev_1_ as a reference. Moreover, cycling vibration loading can cause particle breakage and abrasion. The second/first modulus ratio at load bearing test (k) below 2.2. is established for the calibration of fine soils, which are very different from random fill, where the use of other parameters, such as wheel-tracking and plate bearing tests, prove more useful. The second modulus provides no information on the degree of compaction, so that other criteria based on the first modulus are considered more appropriate. This underdevelopment suggests the need for a new compaction control procedure, which entails the need for different functional parameters, such as automatic online complete process monitoring or specific loading plate diameters. The strongly anisotropic properties of slate make it suitable for its use in random fill or sub-base layers for any traffic volume.

## 2. Materials and Methods

The random fill material and field tests were carried out on the Spanish motorway A-66 "Ruta de la Plata”.

This research focuses on slate rocks. The reason for choosing the slate family is that these rocks are usually obtained from highway diggings, similar to those obtained from the demolition of buildings or pavements. Thus, the scientific knowledge was focused on slates because is a rock that is widely abundant and, therefore, the results on this material have wide applicability. Table 2 provides a summary, including examples of the tests that were conducted on the slate alluvial material during excavation, with the last row showing average values.

The technical standard requirements for the slate alluvial materials used in the laboratory experiments were the particle size by screening, UNE 103101. [15]; determination of the liquid limit and plastic limit of a soil, UNE 103103 [16] and UNE 103104 [17]; modified Proctor compaction test UNE 103501 [18] and California Bearing Ratio (CBR), UNE 103502 [19].

These soils come from the alteration of slates and are associated with low to medium plasticity. According to the Unified Soil Classification System (USCS), most of them belong to the GC group of the coarse-grained soils wrapped in a clay matrix. Large sizes of the parent rock remain, the percentage after sifting through a 20 mm sieve being 68.8%, and, at the same time, there is an important percentage of fine fractions, with an average of 29.3% after using a fine sieve (0.075 mm). Bedrock weathering variations resulted in the classification of a significant number of samples within the group of high plasticity silts (MH). The existence of coarse sizes implies that CBR testing yielded high values, with an average of 20.9.

In this research, the modified compaction control tests according to Teijón-López-Zuazo et al. [20] have been used, which modify the test procedures in the wheel-tracking test and in the topographic settlements. The study is particularized to the random filling of the crown with slate rocks. To facilitate interpretation, the crown includes the two upper layers of the filling, with thicknesses in the penultimate layer between 60 and 40 cm in the last. All the tests that were used in the experiment are shown in Table 3.

X-ray powder diffraction (XRD) is an analytical technique that we are not choosing to perform because it does not have a strong relation with the compaction quality control, which is strongly associated to different characteristics like mechanical properties.

The wheel-tracking test is carried out with a metal structure on which it is measured. These are welded profiles known as "H" shapes. The wheel-tracking test provides the measurement points in compaction batches, which are between 100 and 200 m. The truck should be conducted through topographic leveling pegs. The test result is the average value between different of measurements before and after the passage of the truck. The pegs reduce the possibility of extreme erroneous observations and the chance of any potential errors.

The other trial reviewed is the topographical settlement. It measures the seats after roller passes. This control method and its limitations were thoroughly revised in the research [19]. The criteria suggested for quality control in the crown is grouped in Table 4.

The degree of compaction proposed is associated with a modified Proctor compaction energy level. All the tests were performed under the same moisture conditions to prevent soil stiffness increases and noticeable dry density decreases in the PLT as a result of decreases in water content to below optimum levels. The ANOVA statistical analysis and Levene’s F test have been done. As a large sample size was obtained, the Kolmogorov–Sminornov test was used to check for normal distribution. Alternatively, the Shapiro–Wilk test was used. When processing road geotechnical tests, a strong association between variables is considered when the value of parameter R2 is higher than 0.70. To summarize, the multivariate analysis ANOVA offered a generalized, single, linear model for the adjustment. There is no difference between dependent and independent indicators with the highest goodness-of-fit.

## 3. Results

Possible linear correlations between two compaction control tests were extensively explored. Test results were gathered in compaction lots, so that the batches with the two analyzed tests were represented as dots. The linear nature of the adjustment studied the definition of the variable as dependent or independent irrelevant. A total of 60 compaction lots were evaluated for possible correlations. There was no relationship between the following tests: density–topographic settlement, wheel-tracking–topographic settlement, density–first PLT modulus (ɸ 600mm) and the wheel-tracking relationship between the second and first modules of the PLT. The uncertainty of measurement data points is a centesimal of a millimeter for the topographic settlement and the wheel-tracking test and 0.1 MPa for the PLT.

### 3.1. Relationship between Topographic Settlement Test and First Modulus PLT (ɸ 600mm)

The topographical settlement test and the first PLT module (ɸ 600mm) have a strong correlation, as can be seen in Figure 1.

Table 5 shows a high value of the Pearson correlation coefficient, ρ = 0.878. There is a low standard error (Se) = 18.1544 MPa. The coefficient of determination validates a variance of 77.2%. All the parameters are indicators of a high correlation between both variables.

ANOVA analysis parameters are in Table 6. Levene’s test is clearly significant, F = 15.315, sig = 0.021. The homoscedasticity criterion is not clearly met. Variances are significantly different since the variables are strongly related.

Table 7 shows high t-values of 5.951 and -3.676, which are both significant. The topographic settlement test predicts the first modulus of PLT (ɸ 600mm)

According to the regression coefficients, the adjustment line is:Ev_1_ = 202.278 – 32.661 s
R^2^ = 0.772
where Ev_1_ is the first module of the PLT (ɸ 600mm) in megapascals and the topographic settlement test in millimeters. The domain of the function uses the intervals of (20 ≤ Ev_1_ ≤ 140) and (3.0 ≤ s ≤ 6.0). The error bars, with standard deviations for 66 measurements (11 per control section), are shown in Figure 2.

### 3.2. Relationship between Wheel-Tracking Test and First Modulus PLT (ɸ 600mm) at Crown

There is a relationship of dependence between both variables. Figure 3 shows the inverse proportionality.

Table 8 shows a high value of the Pearson coefficient, ρ = 0.881, which is associated with low dispersion. The coefficient of determination R^2^ = 0.795 yields a variance of 79.5%. The standard error is only 8.7947 MPa.

The results of the analysis of variance can be seen in Table 9. The high value of F = 46,404 clearly indicates the difference in the variances.

The t-test in Table 10 offers high values, 18.723 and -6.812; both are significant (sig = 0.000).

It can be concluded that both variables, wheel impression test and first modulus of the PLT, are deductible from each other by the following expression:Ev_1_ = 113.937 – 15.932 h
R^2^ = 0.795

The domain of the function lies between the intervals of (30 ≤ Ev_1_ ≤ 110) and (0.0 ≤ h ≤ 5.0). The error bars, with standard deviations for 154 measurements (11 per control section) are shown in Figure 4.

### 3.3. Relationship between Topographic Settlement Test and Second Modulus PLT (ɸ 600mm)

As shown in Figure 5, there is a high correlation between the topographic settlement and the first modulus of the plate bearing test. The distribution is inversely proportional to the lower settlement values, corresponding to the higher values of the second modulus of the plate bearing test (ɸ 600mm).

Table 11 describes a high value of the Pearson correlation coefficient, ρ = 0.993. There is a low standard error (Se) = 2.5483 MPa and a high coefficient of determination R^2^ = 0.986. In other words, there is a high correlation associated with low dispersion.

The ANOVA analysis offers the parameters presented in Table 10. Levene’s test is clearly significant, sig = 0.000 with a value of F = 280.006. Therefore, the assumption of homoscedasticity is not met, since variances are significantly different. The variables have a clear, strong dependency relationship.

Table 12 shows the analysis of variance. There is a high correlation between both variables, as can be seen with the high value of Levene F = 280.006

Student’s t-test values are significant. As shown in Table 13, there is a significant contribution by the topographic settlement in the second modulus of the plate bearing test (ɸ 600mm).

The expression of the adjustment line is:Ev_2_ = 224.455 – 19.725 s
R^2^ = 0.986
where s is the topographic settlement in millimeters and Ev_2_ is the second modulus of the plate bearing test in megapascals. The domain of the function has values between (110 ≤ Ev_2_ ≤ 180) and (2.5 ≤ s ≤ 5.5). The error bars, with standard deviations for 66 measurements (11 per control section), are shown in Figure 6.

### 3.4. Relationship between Wheel-Tracking Test and Second Modulus PLT (ɸ 600mm)

Figure 7 shows the inverse proportionality between the second modulus of PLT (ɸ 600mm) and the wheel impression test.

Table 14 shows a high value of the Pearson correlation coefficient, ρ = 0.854, which is associated with low dispersion. The coefficient of determination R^2^ = 0.729 yields a variance of 72.9%. The standard error is only 15.6612 MPa.

The ANOVA analysis offers the parameters listed in Table 15. Levene’s test was significant (sig = 0.000) with a value of F = 26.900. Therefore, the null hypothesis of homoscedasticity is rejected and variances are significantly different.

The t-test in Table 16 offers high values, 18.227 and -5.187, both significant (sig = 0.000).

Moreover, the wheel-tracking test predicts the second modulus of the plate bearing test. Along with the linear regression coefficients, the fit between the wheel-tracking test and the second modulus of the plate bearing test (ɸ 600mm) is:Ev_2_ = 209.559 – 22.077 h
R^2^ = 0.729
where Ev_2_ is the second modulus of PLT (ɸ 600mm) in megapascals and h is the wheel-tracking test in millimeters. The domain of the function lies between the intervals of (100 ≤ Ev_2_ ≤ 210) and (0.0 ≤ h ≤ 5.0). The error bars, with standard deviations for 132 measurements (11 per control section), are shown in Figure 8.

### 3.5. Slate Random Fill in Crown Significance Matrix

The results were grouped in Table 17, which shows the combinations of analyzed tests with their corresponding coefficients of determination. Non-representative numerical values have been replaced by ns (not significant). Other results were not included in the significance matrix because they were obvious.

A more complex summary of the statistical analysis is shown in Table 18.

The in situ density test did not correlate with any other variable. According to the results, the two moduli of the plate bearing test (Ev_1_ and Ev_2_) proved to have a strong relationship with both the wheel-tracking and the topographic settlement tests. A revised control method has been designed for the in situ density test and the PLT (ɸ 600mm).

## 4. Discussion

The topographic settlement test usually measures the first and last pass of the compaction roller. In the revised procedure, measurements are also taken on the penultimate and last pass of the compactor. There is a strong correlation of the topographic settlement improved with the PLT (ɸ 600mm), so one of them can easily be deduced from the other. 

The wheel impression test lacks precision. The test distance is only 10 meters and the measurements are made on the ground. The test has been revised to improve on these deficiencies by using metal picks, doubling the number of measurement points and a test distance of 50 meters. As a dependency relationship was found between the revised test and the PLT (ɸ 600mm), this allows the wheel impression test to be replaced by the PLT (ɸ 600mm).

As the maximum dry density is obtained by laboratory compaction using a modified Proctor test, the degree of compaction is obtained from the field of dry density. However, average density control using nuclear methods is characterized by its high heterogeneity, low performance and testing of only low degrees of thickness, making plate bearing tests necessary to assess stiffness. To evaluate the quality of compacted soil only from the results of plate bearing tests, these were performed using compacted soils with a moisture content within a specific interval (−2, +1%) above modified Proctor optimum water content (w_opt_). A decrease in the water content from w_opt_ according to modified Proctor means an increase in stiffness according to PBT, whereas dry density decreases. The PBT is a test where the highest pressure of the load is on the surface, providing surface measurements and strongly associated with surface moisture. Therefore, surface moisture is the main parameter in the result of the test. Due to this, all the PBT were carried out immediately after nuclear tests. In other words, density and PBT were defined at the same moisture content. Hence, the results from the in situ density test and the PLT (ɸ 600mm) provide an evaluation of the quality of compacted soil in terms of the degree of compaction requirements. Additionally, analyzed tests have yielded excellent results, supporting the possibility of using sizes larger than fine grain soils in random fill at the crown level.

## 5. Conclusions

The maximum size of the random fill particles conditions the effectiveness of compaction tests, such as in situ density, modified Proctor, PLT, topographic settlements and wheel-tracking tests. The new procedure revises the wheel-tracking test and the topographical settlement test, optimizing the results. Finally, statistical analysis allows for the simplification of the quality control procedure for random slate fills. The contributions of the research are:A revised procedure of the wheel-tracking test and topographic settlement control method were adapted correctly in the new compaction quality control in core slate random embankments;The in situ density did not correlate with any other variable, limited by particle dimensions and layer thicknesses;The plate bearing test (for ɸ 300mm) has limitations on random embankment quality control. It requires the diameter of the element to be five times the maximum size of the aggregate (400mm).The wheel-tracking test correlates strongly (Pearson correlation coefficients, ρ = 0.795 and 0.729) with the modulus of the plate bearing test (ɸ 600mm) and can therefore be replaced to avoid redundant results, when the wheel-tracking test has values between 0 ≤ h ≤ 5 mm and when the plate test has values between 30 ≤ Ev_1_ ≤ 110 in the first modulus and 100 ≤ Ev_2_ ≤ 210 in the second modulus;For crown slate random fill, there is a high correlation (Pearson correlation coefficients, ρ = 0.782 and 0.986) between the topographic settlement and the modulus of the plate bearing test (ɸ 600mm) and can therefore be replaced to avoid redundant results, when the topographic settlement has values between 2.5 ≤ s ≤ 5.5 mm and when the plate test has values between 20 ≤ Ev_1_ ≤ 140 in the first modulus and 110 ≤ Ev_2_ ≤ 180 in the second modulus;The new methods with improved tests proposed for the quality control of crown random fill quality control are the in situ density test and the plate bearing test (ɸ 600mm);The proposed methods compared with the conventional methods produce the reduction of leveling errors by means of a fixed point, avoiding ground distortion. In addition, the dynamic effects of track are minimized in the wheel-tracking test and in the topographic settlement;This method reduces test times by the substitution of the compaction control procedure, which is associated with improved construction performance.

## Figures and Tables

**Figure 1 materials-13-01139-f001:**
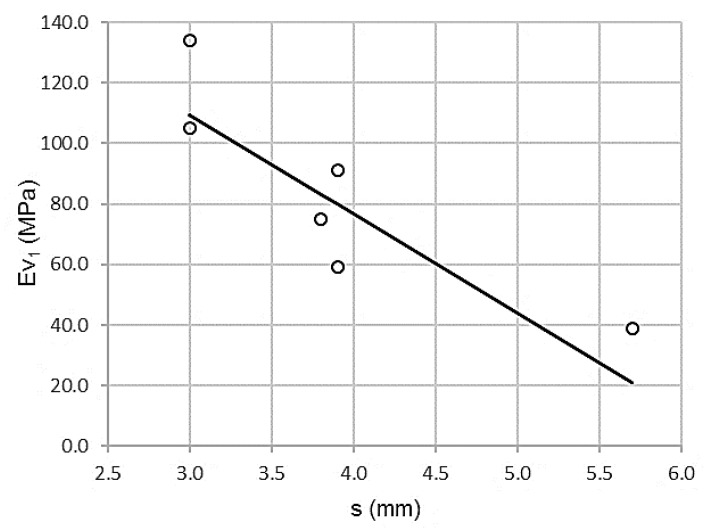
Scatterplot topographic settlement test and first modulus of plate bearing test (PLT) (ɸ 600mm).

**Figure 2 materials-13-01139-f002:**
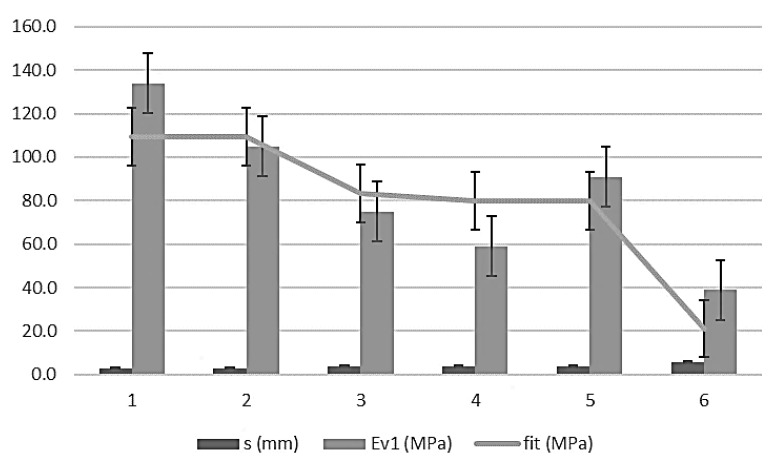
Error bars topographic settlement and first modulus of PLT (ɸ 600mm).

**Figure 3 materials-13-01139-f003:**
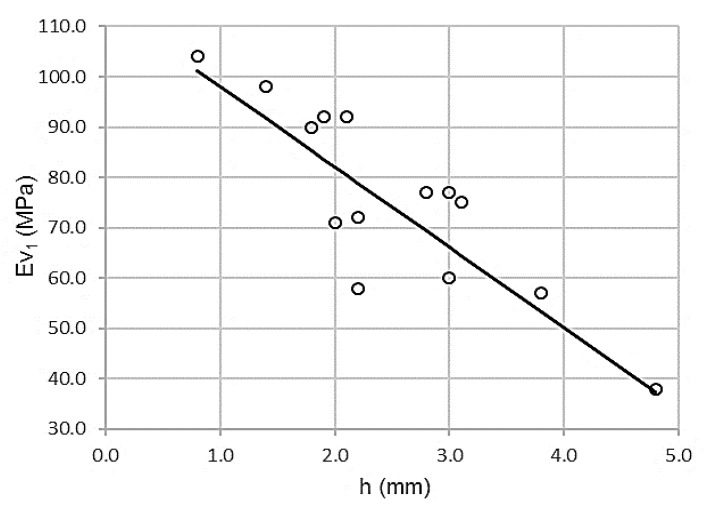
Scatterplot for wheel-track test and first modulus of PLT (ɸ 600mm).

**Figure 4 materials-13-01139-f004:**
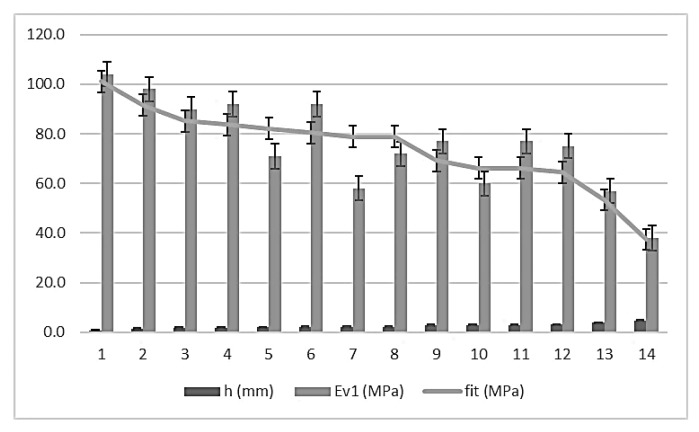
Error bars for topographic settlement and second modulus of PLT (ɸ 600mm).

**Figure 5 materials-13-01139-f005:**
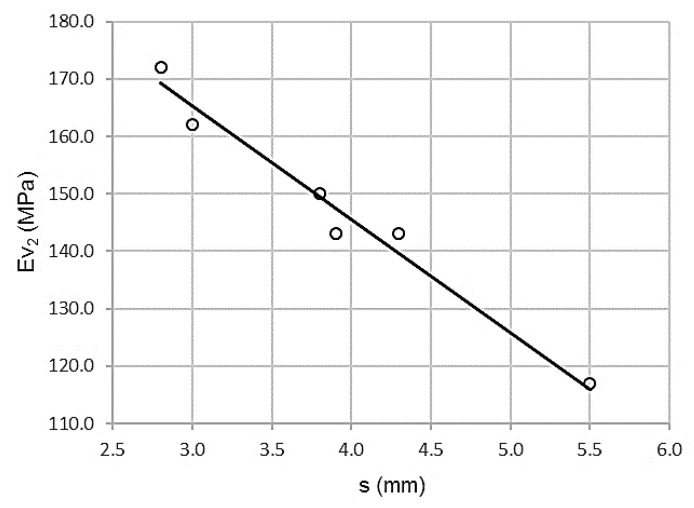
Topographic settlement and second modulus of PLT (ɸ 600mm) at crown.

**Figure 6 materials-13-01139-f006:**
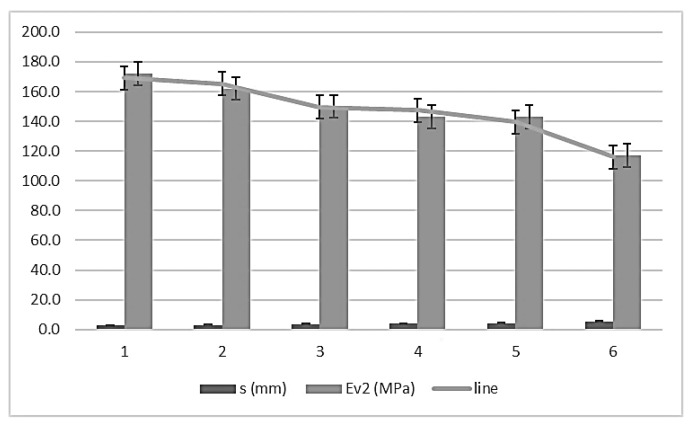
Error bars for topographic settlement and second modulus of PLT (ɸ 600mm).

**Figure 7 materials-13-01139-f007:**
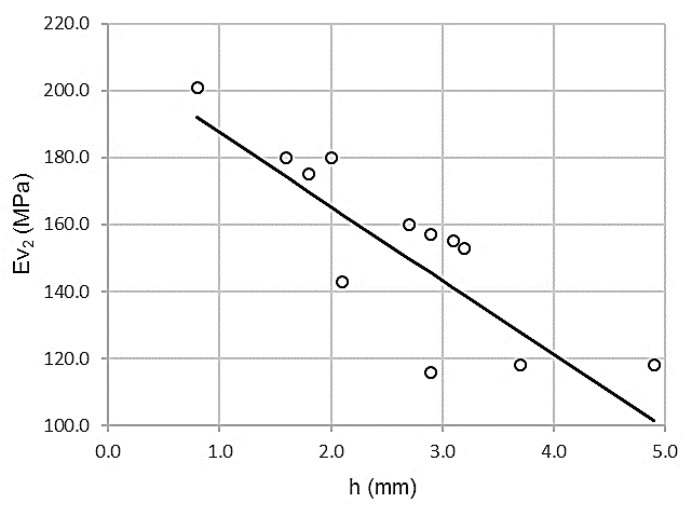
Scatterplot wheel-tracking test and second modulus of PLT (ɸ 600mm).

**Figure 8 materials-13-01139-f008:**
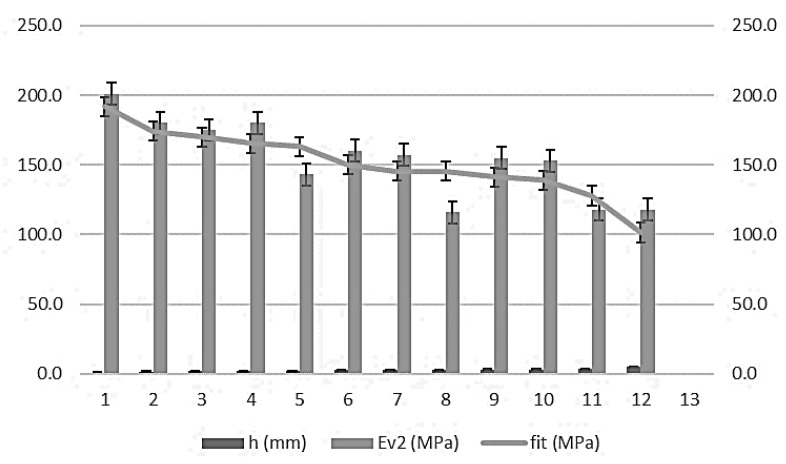
Error bars for wheel-tracking test and second modulus of PLT (ɸ 600mm).

**Table 1 materials-13-01139-t001:** Random embankments quality control.

Test	Research Gaps Based on the Literature Review
topographic control	without reference values
automatic monitoring	strong influence of worker
pit grading	little practical
wheel impression test	usually works
PLT	diameter of the plate five times the maximum size
nuclear density gauging	limited thickness layer
modified Proctor	replacement over 70% original material
sand method	limited size minor than 50mm

**Table 2 materials-13-01139-t002:** Examples of physical parameters for slate alluvial material identification.

Ref.	# 100.0 (mm)	# 20.0 (mm)	# 2.0 (mm)	#0.40 (mm)	#0.075 (mm)	LL	PI	d (g/cm^3^)	H (%)	CBR
CC-016	100.0	51.0	13.0	7.0	4.9	33.1	13.1	2.2	7.0	12.0
CC-013	100.0	68.0	28.0	21.0	17.7	39.0	14.8	2.1	6.1	5.0
CC-010	100.0	52.0	25.0	20.0	14.3	0.0	0.0	2.1	7.9	34.0
I-ELB-1022/04	100.0	75.0	47.0	35.0	23.6	28.0	7.0	2.1	7.7	41.1
1246/04	100.0	100.0	98.0	95.0	94.2	33.5	11.4	1.9	10.5	8.1
1244/04	100.0	67.0	29.0	14.0	8.7	41.9	9.2	2.1	7.3	25.3
Averages	100.0	68.8	40.0	32.0	27.2	29.3	9.3	2.1	7.8	20.9

**Table 3 materials-13-01139-t003:** Compaction quality control methods used in the experiment.

Zone	Tests	Procedure
Laboratory	4500 in situ density and moisture	UNE 103900 [21]
	850 modified Proctor	UNE 103501 [18]
field	960 wheel-tracking tests	UNE 103407 [22]
	580 topographic settlements	PG-3 [23]
	130 plate bearing tests	UNE 103808 [24]

**Table 4 materials-13-01139-t004:** General specifications suggested for crown random fills.

Area	Degree of Compaction (%)	Settlement		Modulus		
		h (mm)	s (mm)	Ev_1_ (MPa)	Ev_2_ (MPa)	k (Ev_2_/Ev_1_)
crown	98.0	≤ 3.0	≤ 4.0	---	≥ 120.0	< 3.6

**Table 5 materials-13-01139-t005:** Determination coefficients for topographic settlement test and first modulus of PLT (ɸ 600mm) at crown.

Summary Model			
R	R^2^	R^2^ adjusted	Standard Error
0.878 ^a^	0.772	0.714	18.1544

^a^ Predictors: constant, s (mm).

**Table 6 materials-13-01139-t006:** Variance analysis for topographic settlement test and first modulus of PLT (ɸ 600mm) at crown.

ANOVA ^a^					
Model	Sum of Squares	Degrees of Freedom	Quadratic Average	F	sig.
regression	4353.665	1	4453.665	13.513	0.021 ^b^
sampling error	1318.329	4	329.582	-	-
total	5571.993	5	-	-	-

^a^ dependent variable: Ev_1_ (mm) ^b^ predictors: (constant), s (mm.)

**Table 7 materials-13-01139-t007:** Linear regression coefficients for topographic settlement test and first modulus of PLT (ɸ 600mm).

Coefficients ^a^					
Model	Nonstandard Coefficients		Standard Coefficients	t	sig.
	B	standard error	beta		
(constant)	208.278	35.001		5.951	0.004
s (mm)	−32.661	8.885	−0.878	−3.676	0.021

^a^ dependent variable: Ev_1_ (MPa).

**Table 8 materials-13-01139-t008:** Determination coefficients for wheel-tracking test and first modulus of PLT (ɸ 600mm) at crown.

Summary model			
R	R^2^	R^2^ Adjusted	Standard Error
0.891 ^a^	0.795	0.777	8.7947

^a^ Predictors: constant, h (mm).

**Table 9 materials-13-01139-t009:** Variance analysis for wheel-tracking test and first modulus of PLT (ɸ 600mm) at crown.

ANOVA ^a^					
Model	Sum of Squares	Degrees of Freedom	Quadratic Average	F	sig.
regression	3589.185	1	3589.185	46.404	0.000 ^b^
sampling error	928.155	12	77.346	-	-
total	4517.340	13	-	-	-

^a^ dependent variable: Ev_1_ (mm) ^b^ predictors: (constant), h (mm).

**Table 10 materials-13-01139-t010:** Linear regression coefficients for wheel-track test and first modulus of PLT (ɸ 600mm) at crown.

Coefficients ^a^					
model	Nonstandard Coefficients		Standard Coefficients	t	sig,
	B	Standard Error	beta		
(constant)	113.937	6.085	-	18.723	0.000
h (mm)	−15.932	2.339	−0.891	−6.812	0.000

^a^ dependent variable: Ev_1_ (MPa).

**Table 11 materials-13-01139-t011:** Determination coefficients for topographic settlement and second modulus of PLT (ɸ 600mm) at crown.

Summary Model			
R	R^2^	R^2^ Adjusted	Standard Error
0.993 ^a^	0.986	0.982	2.5483

^a^ Predictors: constant, s (mm).

**Table 12 materials-13-01139-t012:** Variance analysis for topographic settlement test and second modulus of PLT (ɸ 600mm) at crown.

ANOVA ^a^					
Model	Sum of Squares	Degrees of Freedom	Quadratic Average	F	sig.
regression	1818.352	1	1818.352	280.006	0.000 ^b^
sampling error	25.976	4	6.494	-	-
total	1844.328	5	-	-	-

^a^ dependent variable: Ev_2_ (mm) ^b^ predictors: (constant), s (mm).

**Table 13 materials-13-01139-t013:** Linear regression coefficients for wheel-tracking test and first modulus of PLT (ɸ 600mm) at crown.

Coefficients ^a^					
Model	Nonstandard Coefficients		Standard Coefficients	t	Sig.
	B	Standard Error	beta		
(constant)	224.455	4.675	-	48.009	0.000
s (mm)	−19.725	1.179	−0.993	−16.733	0.000

^a^ dependent variable: Ev_2_ (MPa).

**Table 14 materials-13-01139-t014:** Determination coefficients for wheel-tracking test and second modulus of PLT (ɸ 600mm) at crown.

Summary Model			
R	R^2^	R^2^ Adjusted	Standard Error
0.854 ^a^	0.729	0.702	15.6612

^a^ Predictors: constant, h (mm).

**Table 15 materials-13-01139-t015:** Variance analysis for wheel-tracking test and second modulus of PLT (ɸ 600mm) at crown.

ANOVA ^a^					
Model	Sum of Squares	Degrees of Freedom	Quadratic Average	F	sig.
regression	6597.881	1	6597.881	26.900	0.000 ^b^
sampling error	2452.746	10	245.275		
total	9050.627	11			

^a^ dependent variable: Ev_2_ (mm), ^b^ predictors: (constant), h (mm).

**Table 16 materials-13-01139-t016:** Linear regression coefficients for wheel-tracking test and second modulus of PLT (ɸ 600mm) at crown.

Coefficients ^a^					
Model	Non Standard Coefficients		Standard Coefficients	t	Sig.
	B	Standard Error	beta		
(constant)	209.559	11.497		18.227	0.000
h (mm)	−22.077	4.257	−0.854	−5.187	0.000

^a^ dependent variable: Ev_2_ (MPa).

**Table 17 materials-13-01139-t017:** Slate random fill in crown significance matrix.

Determination Coefficients (R^2^)						
Var	d (g/cm^3^)	h (mm)	s (mm)	Ev_1_ (MPa)	Ev_2_ (MPa)	k (Ev_2_/Ev_1_)
s (mm)	ns	ns	-	-	-	-
Ev_1_ (MPa)	ns	0.795	0.782	-	-	-
Ev_2_ (MPa)	(*)	0.729	0.986	ns	-	-
k (Ev_2_/Ev_1_)	ns	ns	(*)	-	-	-

Nonsignificant (ns), obvious relations (*).

**Table 18 materials-13-01139-t018:** Slate random fill in crown significance matrix.

Student t test (t)						
Var	d (g/cm^3^)	h (mm)	s (mm)	Ev_1_ (MPa)	Ev_2_ (MPa)	k (Ev_2_/Ev_1_)
s (mm)	ns	ns	-	-	-	-
Ev_1_ (MPa)	ns	−6.812	−3.676	-	-	-
Ev_2_ (MPa)	(*)	−5.187	−16.733	ns	-	-
k (Ev_2_/Ev_1_)	ns	ns	(*)	-	-	-

ns: nonsignificant (*) obvious relations.

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
