# Peer review of "Modification Tests to Optimize Highway Construction in Crown of Slate Random Embankments with Compaction Quality Control"

_materials, 2020, doi:10.3390/ma13051139_

Round 1

Reviewer 1 Report

In this study, updated wheel-tracking test and topographic settlement test were developed and applied. The authors collected a large amount of data for performance evaluation. The topic is fascinating and meaningful for the application in civil engineering. However, there are few technical comments to the authors in order to make the paper more readable for the potential readers of the journal. Technical comments: (1) The literature review is useful to present the previous work in the proposed topic in the research area; however, it is necessary to add the respective statements of the research gaps based on the literature review. The research gaps are critical to helping the audiences to understand the study. (2) The respective statements or citations of the technical standard requirements for the used materials in the lab experiments should be included. Although the authors provided some useful lab results to support the proposed topic, in order to make the paper more comfortable to be understood, these technical standard based index analysis is necessary. (3) The authors present a wide range of tests and results, however, the conclusion section does not mentioned many of them. Author should consider to extend further this section.

Author Response

Dear Reviewer:

Thank you very much for your attention. We are pleased to include a revision with the improvement of the technical part. We include 2 archives: the first manuscript materials-717206(3) with only control changes and the second manuscript materials-717206(3)c with changes shown in different color fonts (green for Reviewer 1, yellow for Reviewer 2, blue for Reviewer 3 and red for Reviewer 4). Finally, we attach a PDF with answers to the comments and suggestions.

The authors express their gratitude to the anonymous reviewer who have greatly improved the quality of this manuscript.

Kind regards

Reviewer 2 Report

Attached are the comments

Author Response

(The authors gave the same response as above.)

Reviewer 3 Report

In their manuscript "Modification Tests to Optimize the Highway Construction in Crown of Slate Random Embankments with the Compaction Quality Control", the authors provide a statistically based assessment of the performance of various compaction control tests.
I do not have major concerns on this manuscript, which appears to be rather well written and contains convincing results.
I think the manuscript could benefit of some editorial improvement, though.

For instance, the introduction seems too long and out of focus. I suggest that the authors shorten it, eliminate the subheadings, and focus on the following points: what is the state of the art in compaction control tests, what is the research gap that the authors identified, what motivates them to address it (e.g., how would addressing the gap benefit the scientific knowledge and/or the effectiveness of technical procedures and/or safety of the man made work), how do they plan to address it and why, how is their approach novel compared to other approaches in the literature.

Furthermore, the authors should explain why they focus on a specific material: is this material widely abundant and therefore the results on this material have wide applicability? Or, on the contrary, is this a peculiar and problematic material and why? Are the results on this material extendable in some way to other materials? Answering these questions in the manuscript will clarify the broad interest/impact of the authors' work to the readers

As for the results section, I think the use of so many tables can make reading your manuscript more difficult. I would suggest that the authors merge the ANOVA tables into one and/or find a graphic way to present the results so that they can be compared easily.

Author Response

(The authors gave the same response as above.)

Reviewer 4 Report

This work can be accepted if the following suggestion could be addressed by the authors:

1- There are a few English grammatical errors. Please re-check the manuscript to fix them.

2- the effect of size distribution and particle shape on the compact-ability of granular materials should be more discussed using more references. for example, please see the recent works done by prof. Haddock's (Purdue University) on aggregate structure and compaction of aggregates for asphalt mixtures.

3- improve and strengthen the conclusion part.  

4- Please improve the quality of the figures so as to be more readable. 

Author Response

(The authors gave the same response as above.)

Round 2

Reviewer 2 Report

The authors have done a commendable job to revise the manuscript. Before acceptance, there are a few corrections to suggest:

1) Introduction L-29, "For Teijón el al. [1]" - remove this and add [1] in the end of the sentence

2) L35- Check grammar

3) L65- greywackes, because "of" their

4) L79- Onana et al. [9] research... Research is the wrong tense. Check tense and choose appropriate word

5) L151- that were not - Change were to we are

My only concern is English writing. There are several instances where English correction is required. I strongly suggest proofreading the article with a help of a native speaker.

Author Response

Dear reviewer:

Thank you very much for your attention. We are pleased to envy a new version of our revised manuscript.

The authors express their gratitude to the anonymous reviewer who have greatly improved the quality of this manuscript.

Kind regards

Reviewer 4 Report

This revied manuscript can be accepted for publication.

Author Response

(The authors gave the same response as above.)
